# Numerical Study on Tip Vortex Cavitation Inception on a Foil

**Ilryong Park** [1],*[ID]**, Jein Kim** [1]**, Bugeun Paik** [2] **and Hanshin Seol** [2][ID]

1 Department of Naval Architecture and Ocean Engineering, Dong-Eui University, Busan 47340, Korea; jein@deu.ac.kr

2 Korea Research Institute of Ships & Ocean Engineering, Daejeon 34103, Korea; ppaik@kriso.re.kr (B.P.); seol@kriso.re.kr (H.S.)

* Correspondence: irpark@deu.ac.kr; Tel.: +82-051-890-2595

**Abstract:** In this paper, the inception of tip vortex cavitation in weak water has been predicted using a numerical simulation, and a new scaling concept with variable exponent has also been suggested for cavitation inception index. The numerical simulations of the cavitating flows over an elliptic planform hydrofoil were performed by using the RANS approach with a Eulerian cavitation model. To ensure the accuracy of the present simulations, the effects of the turbulence model and grid resolution on the tip vortex flows were investigated. The turbulence models behaved differently in the boundary layer of the tip region where the tip vortex is developed, which resulted in different pressure and velocity fields in the vortex region. Furthermore, the Reynolds stress model for the finest grid showed a better agreement with the experimental data. The tip vortex cavitation inception numbers for the foil, predicted by using both wetted and cavitating flow simulation approaches, were compared with the measured cavitation index values, showing a good correlation. The current cavitation scaling study also suggested new empirical relations as a function of the Reynolds number substitutable for the two classic constant scaling exponents. This scaling concept showed how the scaling law changes with the Reynolds number and provided a proper scaling value for any given Reynolds numbers under turbulent flow conditions.

**Keywords:** tip vortex cavitation; RANS; Reynolds stress model; cavitation inception; scaling law; elliptic planform foil

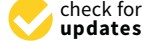



## 1. Introduction

Experimental and numerical analyses of vortex cavitation on hydrofoil-shaped control fins and energy saving devices, rudders and propellers are of crucial importance in naval hydrodynamic problems since this type of cavitation often occurs first and leads to performance deterioration, cavitation erosion damages and cavitation induced noise. The essence of the analyses should be the proper understanding and modeling of the detailed flow physics into the vortex cavitation inception process of a vortex flow. In addition, one of the practical purposes of cavitation analysis is the upscaling of vortex cavitation inception from the model scale data to full scale designs. Although experiments and numerical analyses have their own shortcomings, mutually complementary studies are needed to more accurately and effectively solve cavitation scaling problems. Experimental approaches, analytical methods and numerical schemes for the investigation of the complicated physics involved in cavitating flows have been developed, and these have produced an enormous amount of data on the subject of cavitation. Representative reviews for the subject can be found in Plesset and Posperetti [1], Arndt [2,3], Rood [4] and Arndt [5].

The mechanism of development of tip vortex flow and the appearance characteristics of tip vortex cavitation were examined and reported by Souders and Platzer [6], Arndt et al. [7], Arndt and Keller [8] and Maines and Arndt [9]. The complicatedness of tip vortex flows is made known to be brought about by the turbulence and sharp changes of the pressure and velocity across the vortex core [10]. Several observations in relation to

cavitation in a tip vortex trailing from an elliptic hydrofoil can be found in Arndt et al. [7]. This experimental study shows the dependency of the cavitation number associated with tip vortex cavitation inception on the Reynolds number and gas content value and detailed LDV data of the velocity field inside the tip vortex. As for the detailed mechanism of vortex cavitation inception, the role of nuclei reaching the vortex core as a trigger of cavitation occurrence was investigated by Boulon et al. [11], and afterward, the sensitivity of the tip vortex cavitation developing on a blade to the nuclei distribution was found by Pennings et al. [12].

The cavitation inception point is usually determined by visual observations of cavitation appearance [8]. Recently, acoustic measurement schemes have been utilized as another alternative to detect the cavitation inception by using a signal analysis [13–15]. However, as discussed in Asnaghi et al. [16], since all relevant flow features cannot be measured in the experimental approaches because of small scales of flow dynamics, numerical simulations can be used as a complementary tool to obtain further insights into tip vortex properties [17–19]. It should be noted that it is still a challenging problem for the experimental methods as well as the numerical methods to accurately predict the vortex flows getting complicated by the turbulence, steep pressure and velocity gradients. These flows become more complicated by the presence of phase change [20,21]. The numerical methods to predict the cavitation inception are well classified and explained in Asnaghi et al. [16]. The wetted flow approach without considering cavitation occurrence utilizes the minimum pressure to determine the cavitation inception point, in which cavitation occurrence is expected when the minimum pressure reaches the saturation pressure. The empirical relations obtained from the experimental data showing the dependency of the tip vortex cavitation on the Reynolds number can also be used with the wetted flow simulations [5,8,22]. As a direct simulation approach, in the Eulerian cavitation simulation approach, the cavitation inception number is determined when the computed vapor volume shows similar with a detectable minimum vapor volume at the cavitation inception point [16,18,19,23]. The Lagrangian bubble dynamics approach, which is a somewhat costly method because of the requirement of Lagrangian equations of bubble motions, can determine the cavitation inception point by considering the nuclei effects and the interaction of the tip vortex and nuclei [20,24–27].

In the present study, the inception of tip vortex cavitation for an elliptic planform hydrofoil has been estimated using a numerical simulation and validation tests. In addition, new scaling laws with variable scaling exponent have also been suggested for cavitation inception index. Numerical investigation of the fully wetted and cavitating flows around the hydrofoil was carried out. Reynolds averaged Navier–Stokes (RANS) approach along with an Eulerian cavitation model was used for the vortex cavitation flow simulations. The validation work evaluated the effects of the turbulence model and grid resolution on the tip vortex flows. Some past studies reported that the RANS methods have been known to be inadequate in predicting the tip vortex flow accurately [18,28–31]. This was because the one-equation and two-equation turbulence models used in the RANS simulations showed excessive numerical diffusion inside the tip vortex [18,23,24]. In this regard, the current study confirms that the RANS method using the Reynolds stress model (RSM) can achieve an accurate representation of this kind of tip vortex flows and can adequately explain the flow physics involved.

In the scaling of tip vortex cavitation inception, the extrapolation of the cavitation index from the model tests to large scale conditions is generally overestimated when the classical scaling laws are used [24,32]. The McCormick's law [22] and Arndt's law [5,8] are based on the rule that the cavitation inception is proportional to the Reynolds number raised to an exponent, $R_e^\gamma$. The recent analytical studies using the McCormick's law under limited Reynolds number conditions have suggested various values lower than the classical value of the exponent $\gamma$ [24,33–35]. However, the current cavitation scaling study suggests a new expression of the scaling exponent $\gamma$ as a function of the Reynolds number and

shows the variation of the suggested scaling exponent at higher Reynolds numbers for full scale applications.

The current numerical results are overall divided into two parts. The first part is about the validation of the present numerical solutions for the wetted and cavitating flows. In the second part, the results of the tip vortex cavitation inception analysis on an elliptic planform foil are presented and discussed under the light of the present new scaling concept.

## 2. Numerical Approaches

### 2.1. Governing Equations of Flows

The tip vortex flows around an elliptic hydrofoil were described by the RANS equations in which the flow quantities are split into an average and a fluctuating part. They can be derived from mass and momentum conservation principles based on the assumption of continuum mechanics. The continuity and momentum equations for two-phase incompressible flows in differential conservation form can be written as follows:

$$\frac{\partial \overline{u}_i}{\partial x_j} = 0 \tag{1}$$

$$\frac{\partial (\rho \overline{u}_i)}{\partial t} + \frac{\partial \left(\rho \overline{u}_i \overline{u}_j\right)}{\partial x_j} = -\frac{\partial \overline{p}}{\partial x_i} + \frac{\partial}{\partial x_j}\left(\mu \frac{\partial \overline{u}_i}{\partial x_j} - \rho \overline{u'_i u'_j}\right) \tag{2}$$

where $\rho$ is the density, $t$ is time, $u_i$ are the velocity components in the direction of the Cartesian coordinates $x_i$, $u'_i$ are the velocity fluctuations, $p$ is the pressure, and $\mu$ is the dynamic viscosity. The unclosed term $\tau_{ij} = \rho \overline{u'_i u'_j}$ is the turbulent Reynolds stress tensor. Most turbulence models rely on the so-called Boussinesq analogy, whereby the anisotropic part of $\tau_{ij}$ is written as a linear function of the mean rate of strain as follows:

$$\tau_{ij} = -\mu_t \left(\frac{\partial \overline{u}_i}{\partial x_j} + \frac{\partial \overline{u}_j}{\partial x_i}\right) + \frac{2k}{3}\delta_{ij} \tag{3}$$

where $k$ is the turbulent kinetic energy, $\frac{2}{3}k\delta_{ij}$ in the isotropic part of the Reynolds stresses, and $\mu_t$ is the turbulent eddy viscosity. On the other hand, the Reynolds stresses can be determined by solving the Reynolds stress transport equations as follows:

$$\begin{aligned}
&\frac{\partial}{\partial t}\left(\rho \overline{u'_i u'_j}\right) + \frac{\partial}{\partial x_k}\left(\rho u_k \overline{u'_i u'_j}\right) = -\frac{\partial}{\partial x_k}\left[\rho \overline{u'_i u'_j u'_k} + \overline{p\left(\delta_{kj}u'_i + \delta_{ik}u'_j\right)}\right] \\
&+ \frac{\partial}{\partial x_k}\left[\mu \frac{\partial}{\partial x_k}\left(\overline{u'_i u'_j}\right)\right] - \rho\left(\overline{u'_i u'_j}\frac{\partial u_j}{\partial x_k} + \overline{u'_j u'_k}\frac{\partial u_i}{\partial x_k}\right) - \rho\beta\left(g_i\overline{u'_j\theta} + g_i\overline{u'_i\theta}\right) \\
&+ \overline{p\left(\frac{\partial u'_i}{\partial x_j} + \frac{\partial u'_j}{\partial x_i}\right)} - 2\mu\overline{\frac{\partial u'_i}{\partial x_k}\frac{\partial u'_j}{\partial x_k}} - 2\rho\Omega_k\left(\overline{u'_j u'_m}\epsilon_{ikm} + \overline{u'_i u'_m}\epsilon_{jkm}\right)
\end{aligned} \tag{4}$$

where $\beta$ is the coefficient of thermal expansion, $g_i$ are the gravity accelerations, and $\Omega_k$ are the rotation rates. The seven terms on the right-hand side of Equation (4) are interpreted as turbulence diffusion, molecular diffusion, stress production, buoyancy production, pressure-strain and dissipation from left to right. This kind of turbulence modeling is referred to as the Reynolds stress model (RSM) which was originally discussed by Launder et al. [36]. Different turbulence models were selected in the simulations of this paper. Among several two-equation models, the realizable *k-ε* model based on the suggestions by Shih et al. [37] and the shear stress transport *k-ω* turbulence model of Menter [38] were used, where abbreviations for the former and the latter model in this paper are RKE and SKW, respectively. The RKE model uses the same turbulence viscosity assumption as the standard *k-ε* model but substantially shows better performances for many applications. The SKW model that blends the *k-ε* model in the outer region and a *k-ω* model in the near wall region to obtain higher accuracy of the *k-ω* model in the near wall region is known to have better ability to capture separation than *k-ε* models. The third turbulence model

selected in this study was the elliptic blending RSM modeling developed by Manceau and Hanjalić [39]. The elliptic blending RSM is a low-Reynolds number model that is based on an inhomogeneous near-wall formulation of the quasi-linear quadratic pressure strain term and uses a blending function to blend the viscous sub-layer and the log-layer formulation of the pressure-strain term. This approach was developed to offer a reasonable comprise between simplicity and consistency with the physics [39].

### 2.2. Schnerr–Sauer Cavitation Model

A homogeneous seed-based approach was used for simulations of cavitating flows. This model, developed by Schnerr and Sauer [40], uses a seed-based mass transfer model for cavitation. The number of seeds in a control volume is assumed to be proportional to the amount of liquid as follows:

$$N = n_o \alpha_l V \tag{5}$$

where $n_o$ is the number of seeds, and $\alpha_l$ is the volume faction of liquid within a control volume $V$ that the vapor and the liquid phases occupy. The total vapor volume within the control volume is defined as follows:

$$V_v = N V_b \tag{6}$$

where $V_b$ is the volume of on bubble. The vapor volume fraction $\alpha_v$ can be expressed as follows:

$$\alpha_v = \frac{4}{3} \pi R^3 n_o \alpha_l \tag{7}$$

where $R$ is the local bubble radius. The vapor bubbles are moving with the flow. The rate at which vapor is developed at a time instant is approximately the rate at which the volume of bubbles present in the control volume at a time instant change. The volume change of an individual cavitation bubble can be expressed as follows:

$$\frac{dV_b}{dt} = 4\pi R^2 V_r \tag{8}$$

where $V_r$ is the bubble growth velocity, $dR/dt$. The Schnerr and Sauer [40] model is based on a reduced Rayleigh–Plesset equation neglecting the effect of bubble growth acceleration, viscous effects and surface tension effects. The cavitation bubble growth rate is approximately computed as follows:

$$V_r^2 = \frac{2}{3} \left( \frac{p_{vap} - p}{\rho_l} \right) \tag{9}$$

where $p_{sat}$ is the vapor pressure corresponding to the temperature at the bubble surface, $p$ is the pressure of the surrounding liquid, and $\rho_l$ is the liquid density. The mass transfer rate per unit volume can be written as

$$\dot{m} = 4\pi n_o \alpha_l \rho_v R^2 V_r \tag{10}$$

where $\rho_v$ is the vapor density. The transport equation for the vapor volume fraction with the mass transfer rate $\dot{m}$ can be expressed as follows:

$$\frac{\partial}{\partial t}(\alpha_v \rho_v) + \frac{\partial}{\partial x_j}(\alpha_v \rho_v u_j) = -\dot{m} \tag{11}$$

### 2.3. Numerical Solution Methods

The governing equations for two-phase incompressible flows were solved in the commercial program Star-CCM+ by using the finite volume method and a volume of fluid approach [41]. The segregated flow approach was used to solve the equations in which the Semi-Implicit Method for Pressure-Linked Equations (SIMPLE) algorithm was adopted

to resolve the pressure-velocity coupling. The spatial differencing of the convective terms uses a second order accurate upwind scheme. A second-order central differencing was used for the viscous terms. For the solutions of non-cavitating tip vortex flows, steady computations were carried out. The cavitating flows that require unsteady simulations with the multiphase volume of fluid method were solved with a second order time implicit scheme and 10 inner iterations to reach convergence for a given physical time, where the converged steady-state solutions for non-cavitating flows were utilized as the initial conditions. The computational time step was set to be $1 \times 10^{-3}$ s. This time scale is small enough for the current problems that cover rather stationary tip vortex flows. Unstructured grid based on a hexahedral mesh topology was used in the present simulations. A prism layer was used to generate orthogonal prismatic cells next to the wall surfaces, in which the height of first cells from the wall was determined so that the dimensionless wall distance $y^+$ nearly equals 1. As it is well known, one of the important parts of numerical analysis of tip vortex flows is to find the vortex path and to establish a proper grid resolution around the vortex. To obtain an appropriate grid resolution at the tip vortex core region, a set of refining control volumes was adopted along the tip vortex, where a proper grid size was determined to capture the small-scale tip vortex.

### 3. Numerical Results and Discussion

*3.1. Flow Condition and Grid Resolution*

An elliptic planform foil with the NACA $66_2$-415 section was tested in the present numerical analysis of tip vortex flows. The foil as shown in Figure 1 had a chord length of 0.081 m and a semi-span of 0.095 m as those used in the experiment [7]. The current simulations were overall separated into two parts. The first part considered the validation of the present numerical solutions for tip vortex flows. Non-cavitating flows were computed for the foil with the effective angle of attack $\alpha_e (= \alpha - \alpha_o) = 12°$ at the Reynolds number of $R_e = 5.2 \times 10^5$, where the corresponding flow speed was $U_o = 6.5$ m/s. It should be noted that the practical angle of attack $\alpha$ was 9.5°, and the zero-lift angle of attack $\alpha_o$ was $-2.5°$ as mentioned in Arndt et al. [7]. The boundary layer characteristics near the foil tip predicted at $R_e = 4.8 \times 10^5$ and $\alpha_e = 13°$ was validated through the experimental result of Maines and Arndt [9]. The simulations of cavitating flows around the foil were carried out at the cavitation numbers ($\sigma$), 0.58 and 1.15, and the Reynolds number of $R_e = 5.3 \times 10^5$, in which the foil had the effective angle of attack, $\alpha_e = 9.5°$. In the second part, the cavitation inception of the foil with the effective angle of attack $\alpha_e = 9.5°$ was investigated with the variation of the Reynolds number and cavitation number, where the practical angle of attack corresponded to $\alpha = 7.0°$. The Reynolds number had the range $5.2 \times 10^4$–$1.2 \times 10^6$. In these computations, the cavitation number was increased from a lower value at which the tip vortex cavitation was clearly shown with a large amount until the cavitation was difficult to find around the tip of the foil at each Reynolds number. The present cavitation inception analyses were compared with the cavitation index measured by Arndt and Keller [8]. In addition, the minimum pressure coefficients of wetted flows were also used to estimate the cavitation inception number as follows:

$$\sigma_i = -C_{p, min} \tag{12}$$

The computational domain of the foil as shown in Figure 2 had the same test section of 0.19 m square cross-section with a length of 1.25 m as the experimental facility. The velocity inlet boundary was located approximately six chord lengths upstream of the foil, and the pressure outlet boundary was placed twelve chord lengths downstream of the foil. The wall boundary condition was applied on the foil and other boundaries.

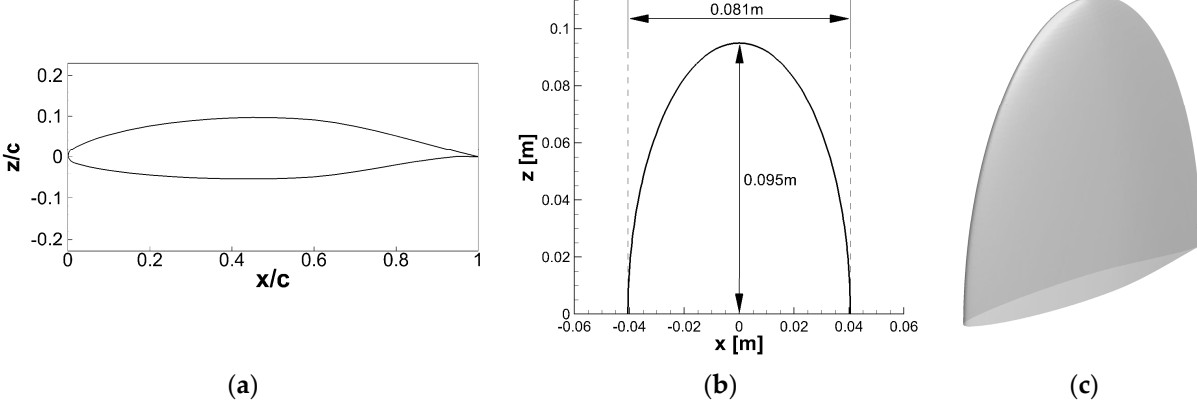

**Figure 1.** Foil section, elliptical profile and perspective view of the elliptic planform foil: (**a**) NACA 66₂-415 section; (**b**) chord and span lengths; (**c**) 3D view of the foil.

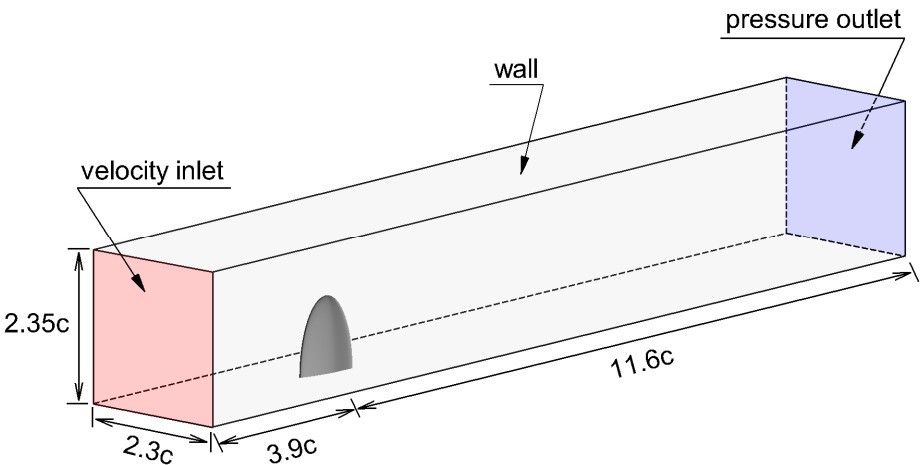

**Figure 2.** Computational domain and boundary conditions.

As briefly mentioned in the previous section, one of the key tasks of numerical analysis of tip vortex flows is to find a proper grid resolution around the tip vortex core. A grid refinement along the vortex core, as shown in Figure 3, was performed after identifying the tip vortex trajectory from the numerical result for an initial coarse grid. A circular cylindrical refinement region of 14 mm diameter was placed along the tip vortex which extended 2.5 times downstream of the foil. The grid distributions on the surface of the foil and in the streamwise and transverse directions are presented in Figure 3. Five different grid resolutions with the dimensionless wall distance, $y^+ \approx 1$ were generated as shown in Table 1. All grids had 20 prism grid layers to have a proper resolution of the boundary layer around the foil. It can be found from the experiment [7] that the tip vortex core at $x/c = 1.0$ had an approximate radius of 1.1 mm for the foil with the effective angle of attack of 12° at a Reynolds number of $5.2 \times 10^5$. The number of grid cells and their sizes across the tip vortex diameter in the grid refinement region are provided in Table 1 for each grid resolution. In the current simulations, 8, 11, 16 and 25 grid cells across the tip vortex core were used for G1, G2, G3 and G4, respectively. Finally, the finest grid resolution, G5, was designed to consist of 28 grid cells across the vortex diameter.

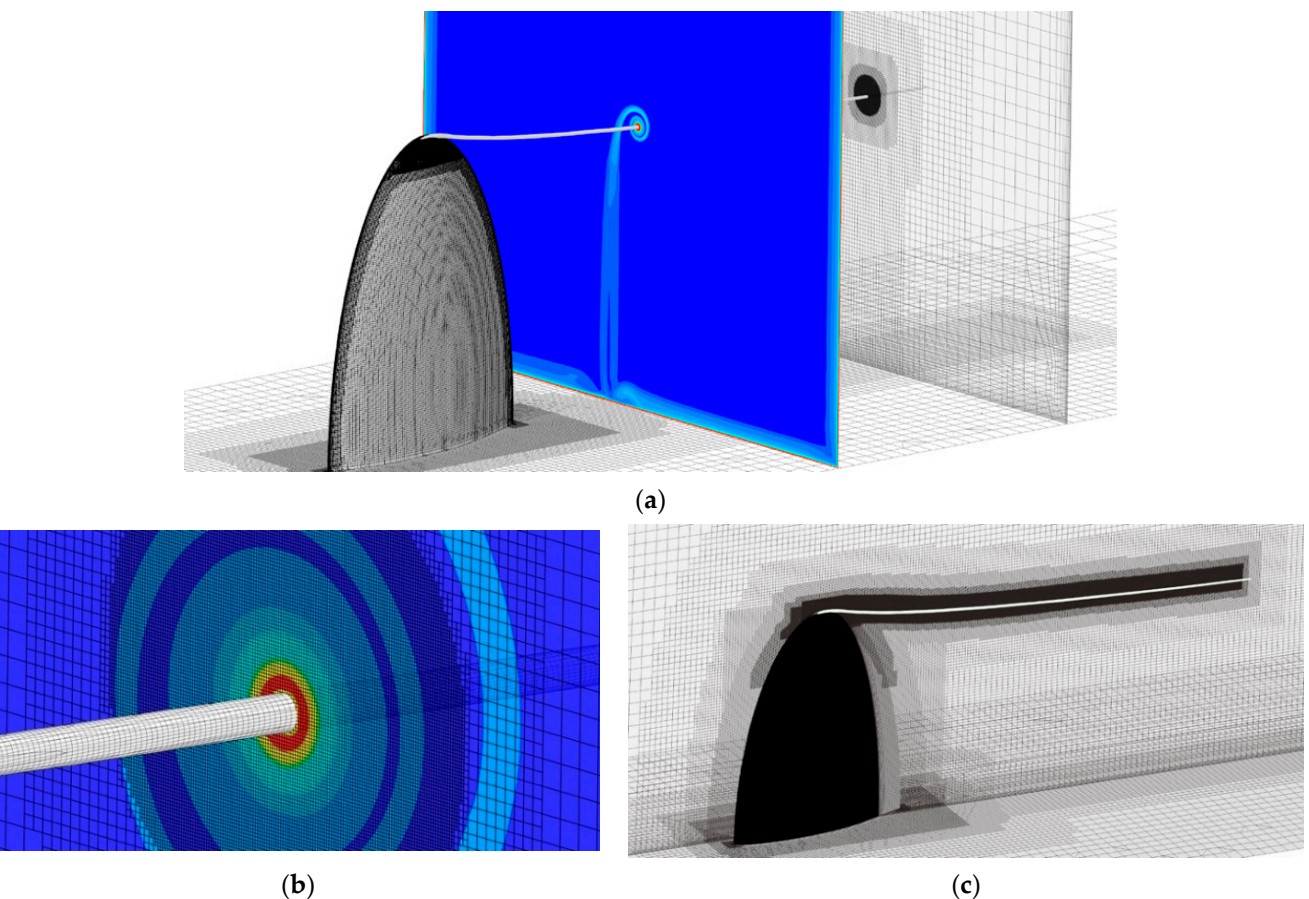

**Figure 3.** Grid distribution on the foil surface and in transverse and streamwise directions: (**a**) surface distribution and transverse distribution; (**b**) close-up view of vortex core region; (**c**) streamwise distribution.

**Table 1.** Computational grid resolutions.

| Grid Name | Total No. of Grids (M) | No. of Grids in Vortex Core | Grid Size (mm) Streamwise Direction | Grid Size (mm) Transverse Direction |
|---|---|---|---|---|
| G1 | 2.5 | 8 | 0.560 | 0.280 |
| G2 | 5.1 | 11 | 0.400 | 0.200 |
| G3 | 10.4 | 16 | 0.280 | 0.140 |
| G4 | 15.7 | 25 | 0.174 | 0.087 |
| G5 | 30.6 | 28 | 0.156 | 0.078 |

*3.2. Validation of Non-Cavitating Flows*

To validate the quality of the present numerical results, flow velocities across the tip vortex core were evaluated. The different grid resolutions were first tested with the RSM turbulence model for numerical evaluations, followed by dependency examination of the turbulence model on the finest grid. Figure 4 compares the vertical velocity profiles extracted along the horizontal line across the tip vortex core measured and calculated at the section of x/c = 1.0 at $R_e$ = 5.2 × 10$^5$ and $\alpha_e$ = 12°. According to the measured results of Arndt et al. [7], the steep gradient of the velocity and its magnitude persisted far downstream of the foil. In these numerical results, it is seen that the influence of grid resolution on the sharpness of the vertical velocity distribution is as large as those of the turbulence model. The SKW and RKE turbulence models underestimated the vertical velocity, resulting in a large difference from the experimental value. On the other hand, the RSM turbulence model shows a better agreement with the experimental data even for

the coarsest grid resolution than the two-equation turbulence models. In the current study, the vertical velocity predicted by the RSM model provided the best agreement with the measured velocity profile for the finest grid resolution G5, followed closely by the next higher grid resolution G4.

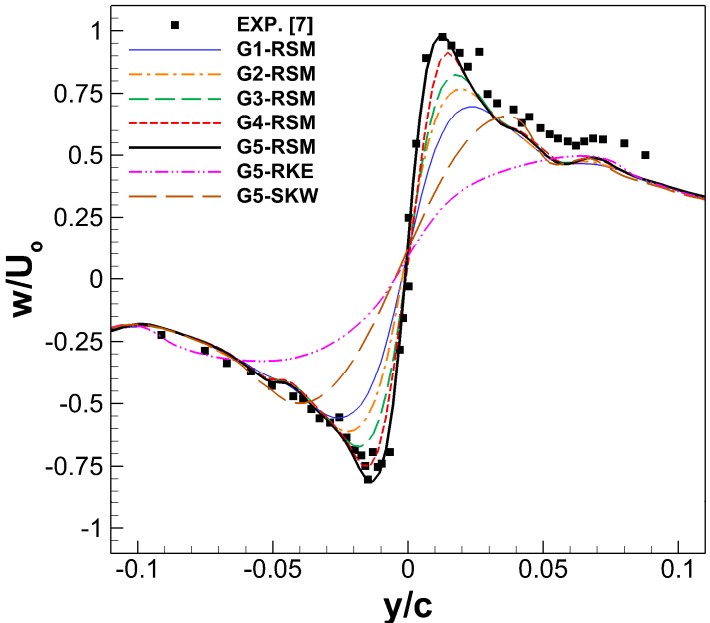

**Figure 4.** Vertical velocity profiles at the section of x/c = 1.0.

The accuracy of the grid resolution G5 in the streamwise direction on the vertical velocity can be found in Figure 5. The streamwise grid size in the tip vortex region is given in Table 1. The magnitude and gradient of the vertical velocity was slightly underpredicted at the further downstream of the foil, x/c = 2.0.

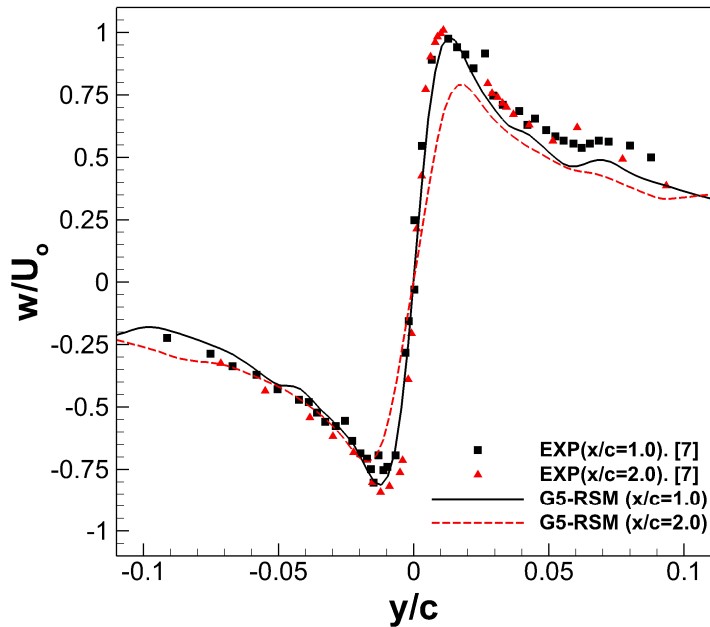

**Figure 5.** Vertical velocity profiles at the sections of x/c = 1.0 and 2.0.

The circumferential velocity profiles calculated under the same flow conditions as in Figure 4 for different grid resolutions and turbulence models were compared with the experimental data in Figure 6. This flow property is a plot of the product of circumferential velocity and radius. The numerical results showed a good prediction of the asymmetry of the tip vortex between the suction and pressure sides. The circumferential velocities predicted by the two-equation turbulence models near the center of the tip vortex core were slightly lower than the measured result.

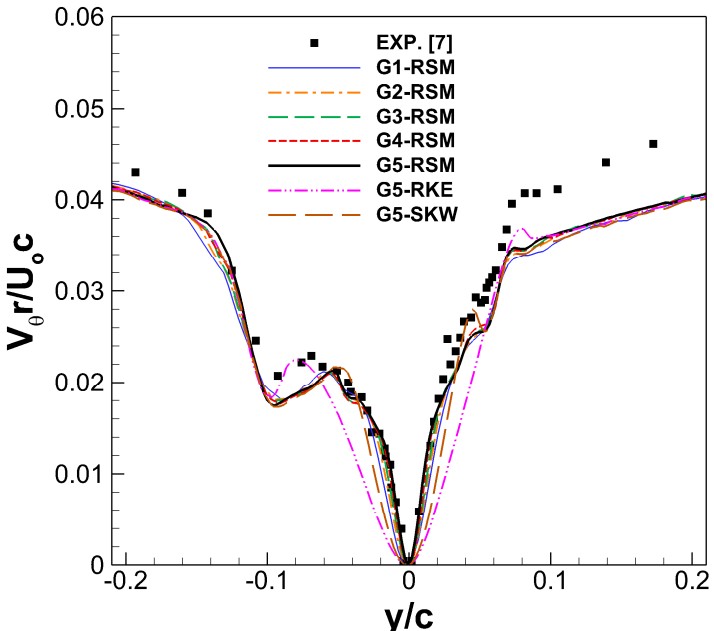

**Figure 6.** Angular momentum profiles at the section of x/c = 1.0.

Figure 7 presents the pressures along the tip vortex core for different grid resolutions and turbulence models. The predicted pressures decreased to minimum values with large changes in slope near the tip of the foil where the initial vortex rollup develops. A grid convergence appeared in the estimation of the minimum pressure, but the influence of the turbulence model was somewhat large in the magnitude and behavior of the pressure coefficient. Moreover, the predicted minimum pressure locations which were very close to the foil tip were found to be almost similar under all numerical conditions, but a change in the pressure downstream of those locations showed different behaviors. The rapid pressure recovery due to the numerical diffusion was seen in the results for the coarser grid resolutions. This numerical trend was more apparent in the pressure variations predicted by the RKE and SKW turbulence modes as previously discussed in Figure 4. This can be attributed to the influence of the increase in numerical diffusion caused by excessive turbulent viscosity.

Figure 8 compares the pressures predicted along the horizontal line across the tip vortex core at x/c = 0.2 and x/c = 1.0. In theory, the pressure distribution can be obtained from the integration of the circumferential velocity shown in Figure 6. The pressure profiles obtained using the RSM model with grid resolutions G4 and G5 showed a sharper variation with similar magnitudes. Even in the results computed by the RKE and SKW turbulence models, the pressure gradients were somewhat large near the tip but decreased significantly due to the increase in the numerical diffusion in the downstream of the foil. From the grid dependence test, the finest grid G5 was considered to be suitable for this study, and the following cavitation analysis was done to investigate the effect of the turbulence model on this grid.

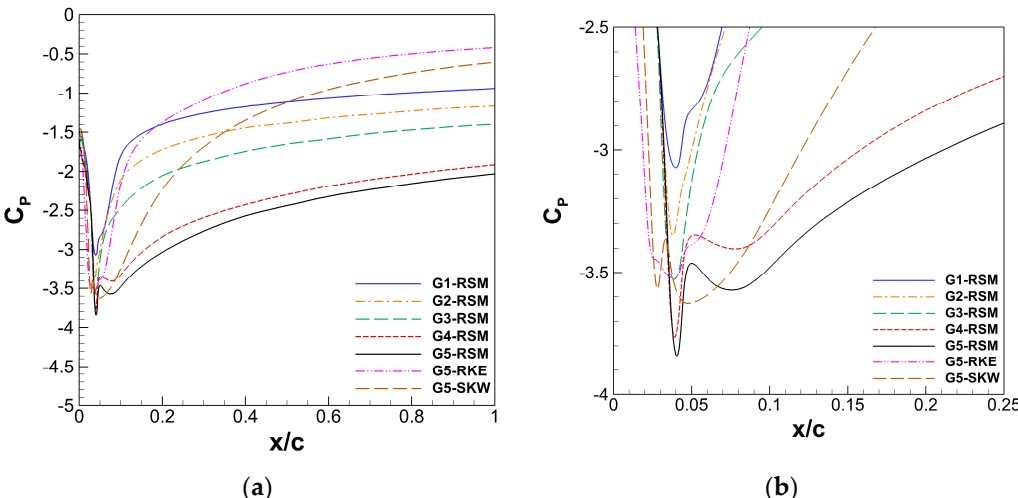

**Figure 7.** Pressure coefficient profiles along the vortex core: (**a**) global; (**b**) close-up view around the minimum pressure.

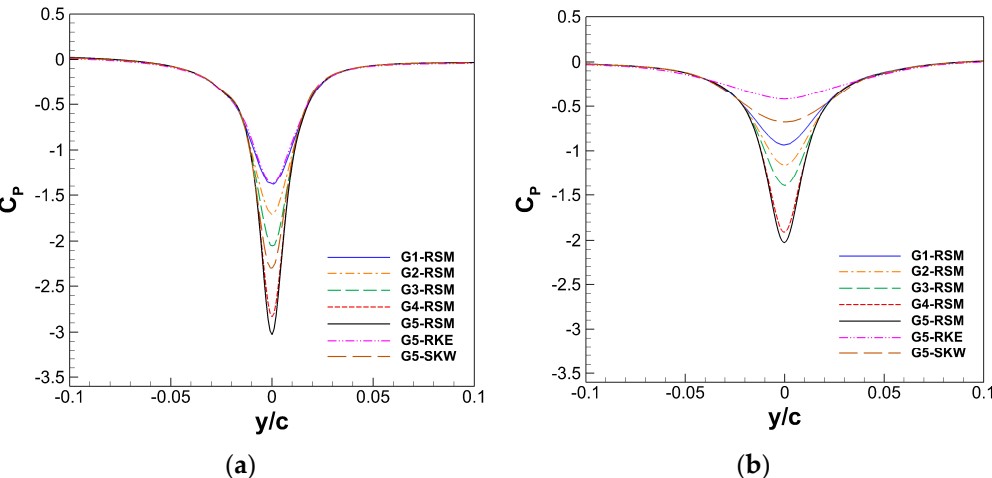

**Figure 8.** Pressure coefficient profiles across the tip vortex core: (**a**) x/c = 0.2; (**b**) x/c = 1.0.

A detailed experimental study of the near tip region was performed for the NACA 16-020, NACA 4215M and NACA $66_2$-415 foils by Maines and Arndt [9]. As an example of their research results, Figure 9a shows a limiting streamline pattern of developed vortex–boundary layer interaction near the tip region of the NACA 16-020 foil suction side. Note that this result is not available for the NACA $66_2$-415. In Figure 9b, the vortex footprints are provided for the NACA 16-020 and NACA $66_2$-415 foils. This vortex footprint clearly defines the separation and reattachment of the vortex flow [9].

In Figure 10, the predicted vortex footprints on the suction surface of the current NACA $66_2$-415 foil are compared with the experimental results, whereas the numerical results on the pressure side are compared with different turbulence models. The simulations were performed for the finest grid G5 at $\alpha_e = 13°$ and $R_e = 4.8 \times 10^5$. The current numerical results showed the limiting streamlines overlapped with the magnitude of the wall shear stress. As discussed in the numerical results for the tip vortex flows, the predicted developed vortex–boundary layer interactions were also different depending upon the turbulence model. The RKE result showed somewhat delayed vortex separation and reattachment, while the SKW result showed an improved agreement with the experimental observation. On the other hand, the RSM provided a better agreement with the measured vortex footprint which can be defined as the area contained between the vortex separation and reattachment lines. When examined together with the wall shear stress shown in

the RSM result, it is seen that the magnitude of this physical quantity is large inside the vortex footprint. Although the suction side boundary layer characteristics varied greatly according to the turbulence model, the pressure side flow features were similar.

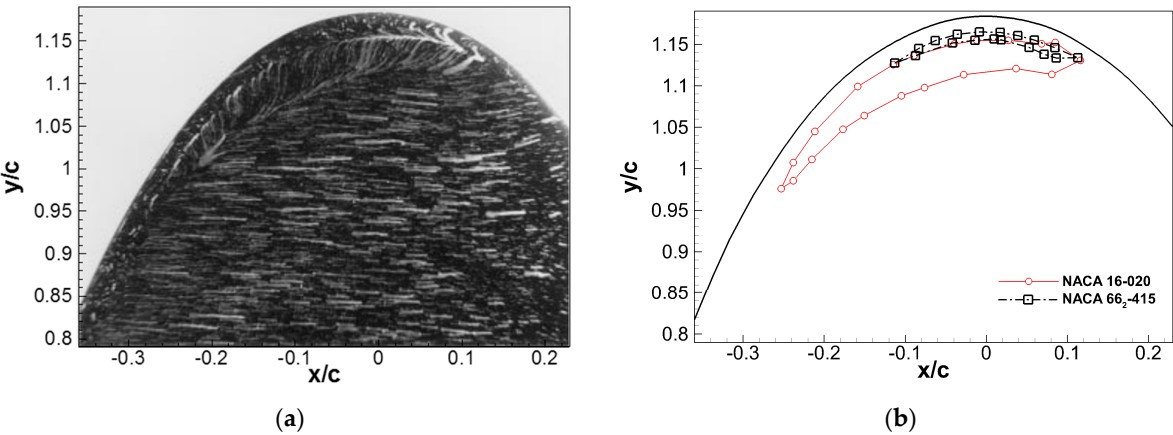

**Figure 9.** Flow details on the suction surface of foils [9]: (**a**) streamline pattern of developed vortex/boundary layer on the NACA 16-020 foil; (**b**) vortex footprints on the NACA 16-020 and NACA $66_2$-415.

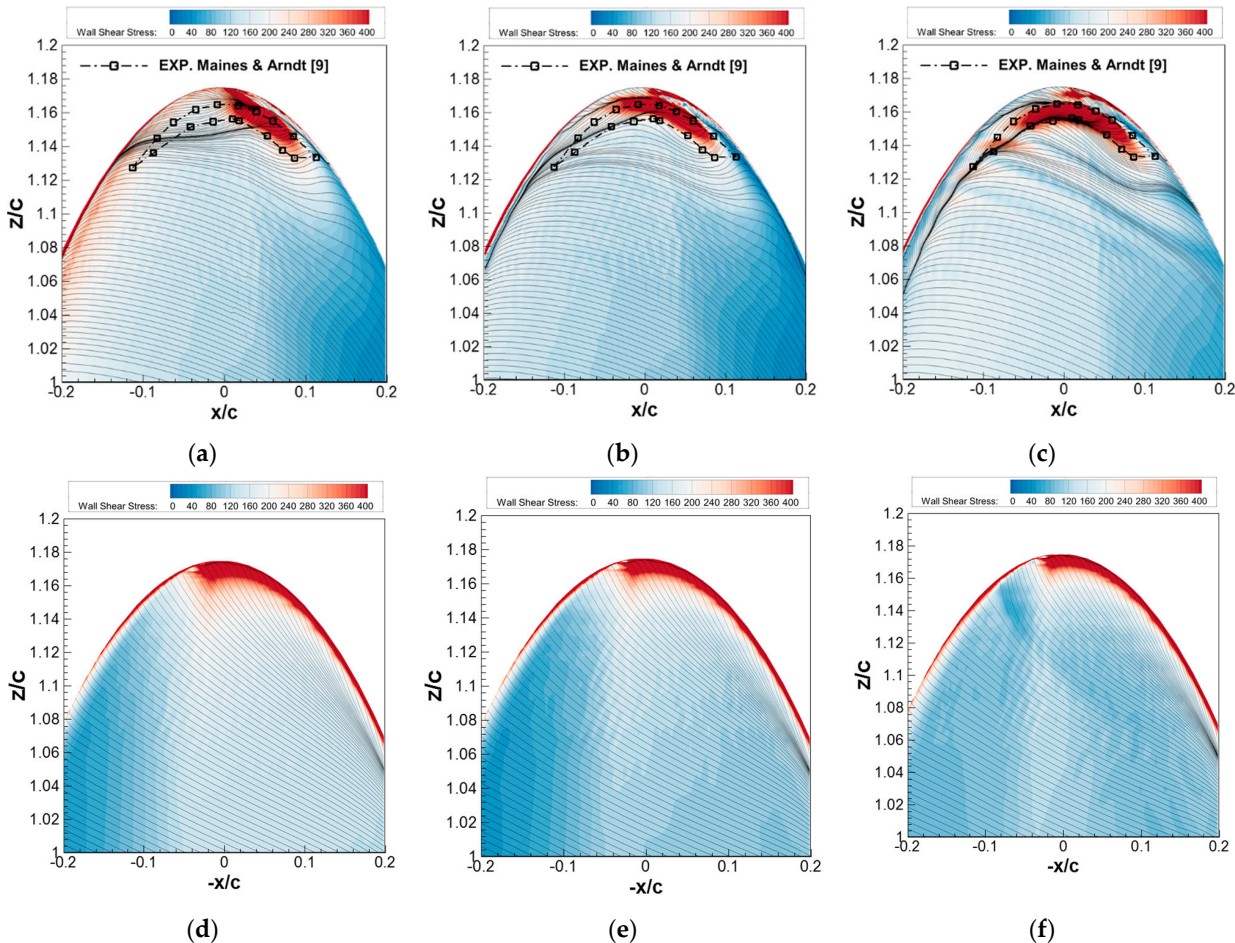

**Figure 10.** Limiting streamlines and wall shear stress distribution on the suction and pressure surfaces of the foil: (**a**) suction side, RKE; (**b**) suction side, SKW; (**c**) suction side, RSM; (**d**) pressure side, RKE; (**e**) pressure side, SKW; (**f**) pressure side, RSM.

Figure 11 compares the pressure distribution on the suction and pressure sides of the foil for different turbulence models. It is seen that for each turbulence model, the low pressure is differently formed within the high wall shear stress region shown in Figure 10. This indicates that the characteristics of cavitation and tip vortex occurrences in the foil can vary to some extent depending upon the turbulence model. Similar to the limiting streamlines and wall shear stress on the pressure side, the pressure distribution on the pressure surface of the foil was also similar for different turbulence models.

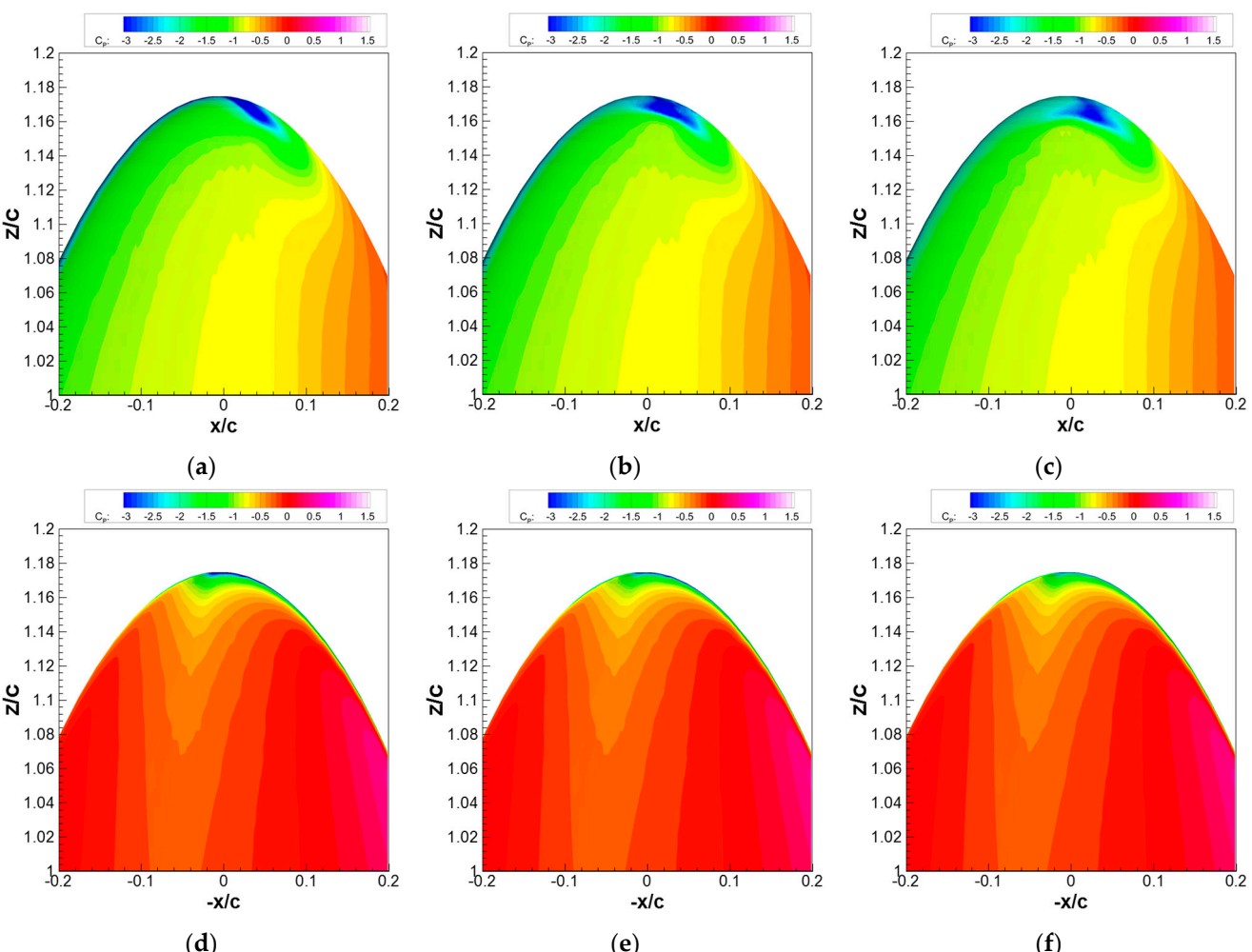

**Figure 11.** Pressure coefficient distribution on the suction and pressure surfaces of the foil: (**a**) suction side, RKE; (**b**) suction side, SKW; (**c**) suction side, RSM; (**d**) pressure side, RKE; (**e**) pressure side, SKW; (**f**) pressure side, RSM.

### 3.3. Validation of Cavitating Flows

This section presents the results of the numerical analysis of the tip vortex cavitation of the target elliptic planform hydrofoil. The cavitating flows around the foil were computed at $R_e = 5.3 \times 10^5$ for the cavitation numbers, $\sigma = 0.58$ and 1.15, in which the foil has an effective angle of attack $\alpha_e = 9.5°$ [7].

Figure 12 shows a comparison of tip vortex cavitation obtained from the experimental observation and the numerical analysis using the RSM turbulence model on the grid resolution G5. The iso-surfaces of the cavitation were plotted as the vapor volume fraction of 0.5. The observed cavitation for the cavitation number of 1.15 in the experiment, attached to the tip, was highly stable both spatially and temporally and extended well into the far downstream of the water tunnel [7]. As discussed by Arndt et al. [7], the tip vortex cavity for the cavitation number of 0.58 showed an instability giving the appearance of a

twisted ribbon. Lastly, the current RSM results for both cavitation numbers showed a good agreement with the cavitation observations of the experiment. In particular, the numerical result for the cavitation number of 0.58 reproduced a similar the stationary wave shape of the cavity, its wavelength, and volume change downstream of the foil. On the other hand, in the current numerical analysis result, it is seen that sheet cavitation covers the surface of the foil. The same trend was found in the numerical results of Frank et al. [23]. This might be due to the surface condition of the foil in the experiment and the nuclei condition in the cavitation tunnel which were not considered in both numerical analyses.

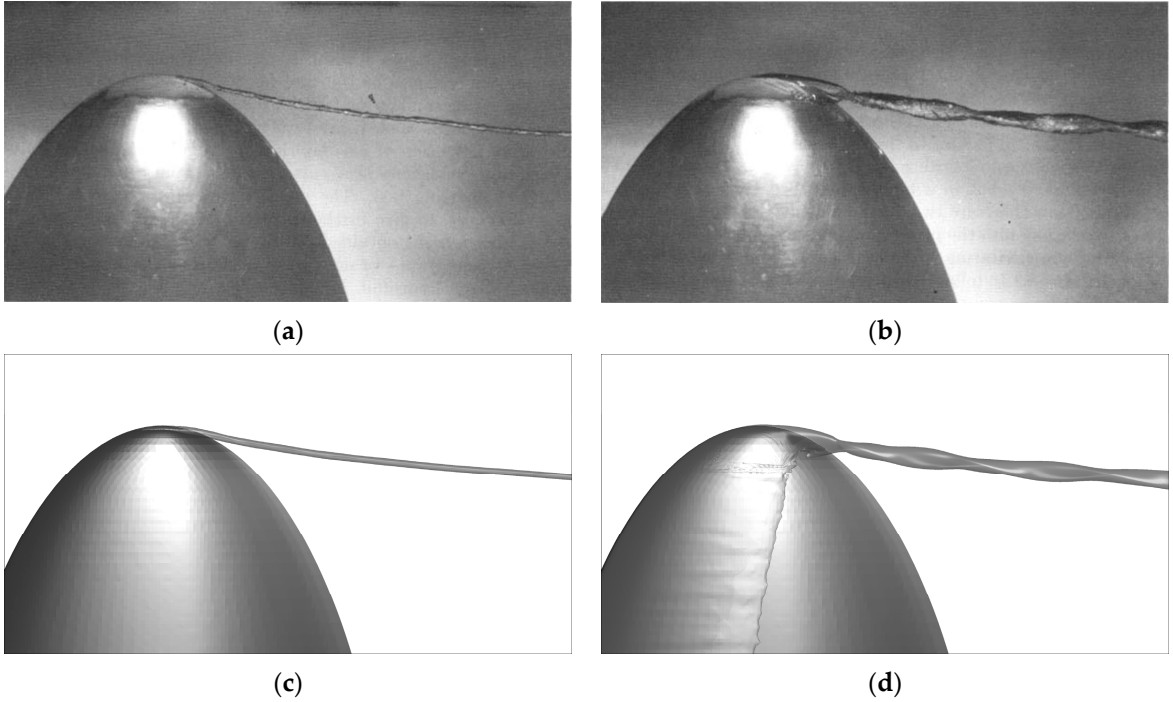

**Figure 12.** Development of tip vortex cavitation at different cavitation numbers: (**a**) EXP, $\sigma$ = 1.15; (**b**) EXP, $\sigma$ = 0.58; (**c**) CFD, $\sigma$ = 1.15; (**d**) CFD, $\sigma$ = 0.58.

The presence of cavitation changed the flow feature in the tip vortex significantly. Figure 13 presents the pressure, axial and vertical velocities and vorticity magnitude at the cross section of the tip vortex at $x/c$ = 1.0 for fully wetted flow and cavitating flow conditions. In Figure 13b,c, the tip vortex cavitation profiles are shown by solid lines. It is seen that the pressure coefficients inside the vortex cavities for each cavitating flow condition are close to the given cavitation numbers. This is due to the low density of the vapor inside the cavity resulting in the reduction in the axial pressure gradient. Furthermore, the axial velocity in the tip vortex region generally decreased due to the decrease in the axial pressure gradient at each cavitation condition, but in the current study, it increased slightly in the vortex cavitation core for the cavitation number of 1.15. The vertical velocity gradient near the cavitation interface was distorted due to the growth of the vapor surrounded by the water as shown in Figure 13b,c. A comparison of vorticity magnitude revealed that the rotation was suppressed inside the vortex cavitation where there were changes in the pressure gradient and velocities due to the density difference between the water and vapor. The influence of cavitation on tip vortex flow is illustrated in Figure 14 where the axial velocity, vertical velocity and pressure coefficient profiles are shown along the horizontal line across the vortex core. As discussed in Figure 13, at lower cavitation number, the axial and vertical velocities greatly varied in the vortex region due to the cavitation growth and reduction in the pressure gradients.

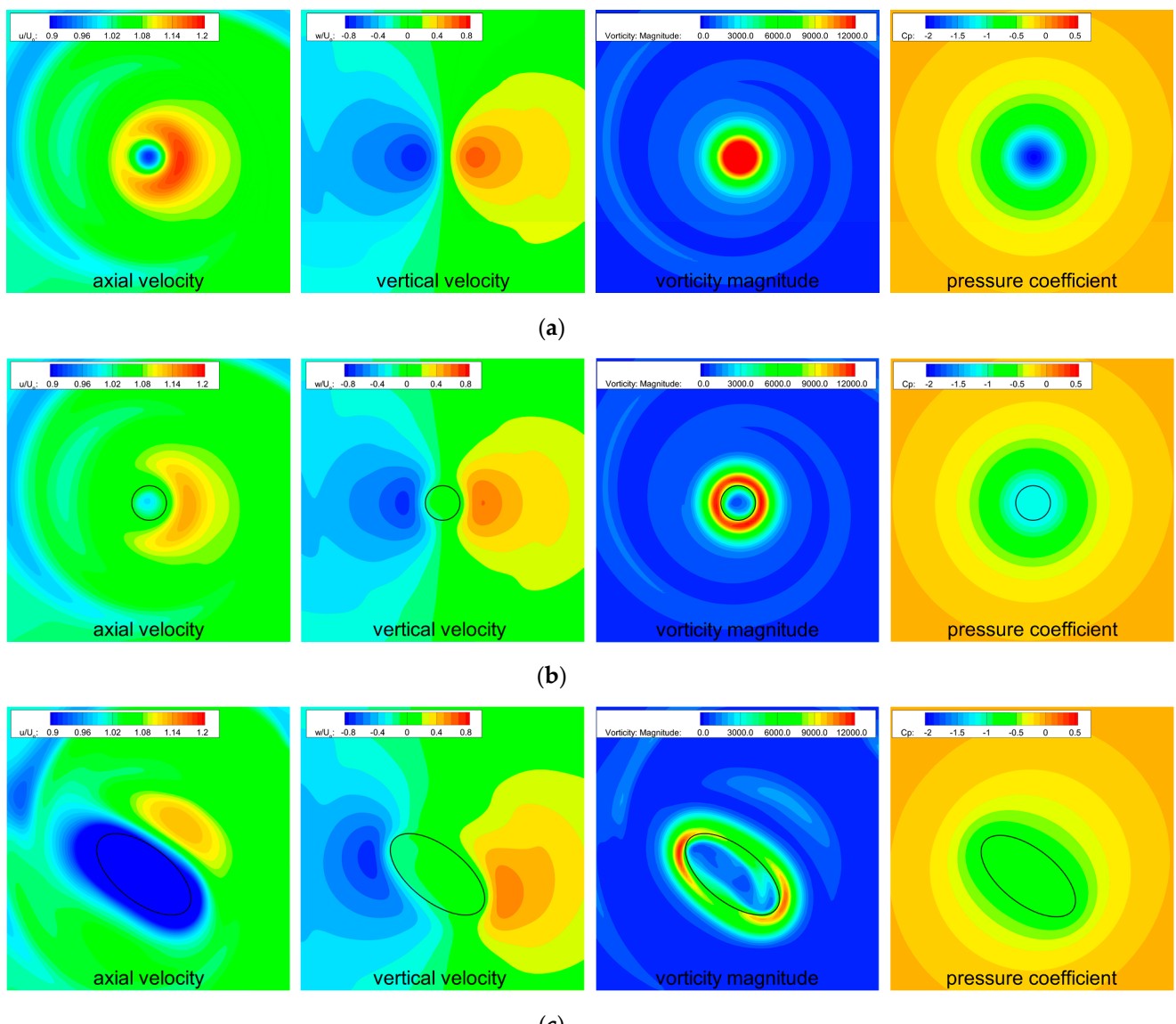

**Figure 13.** Pressure, axial and vertical velocity and vorticity magnitude distributions at x/c = 1.0 for different flow conditions: (**a**) wetted flow condition; (**b**) cavitating flow condition, $\sigma$ = 1.15; (**c**) cavitating flow condition, $\sigma$ = 0.58.

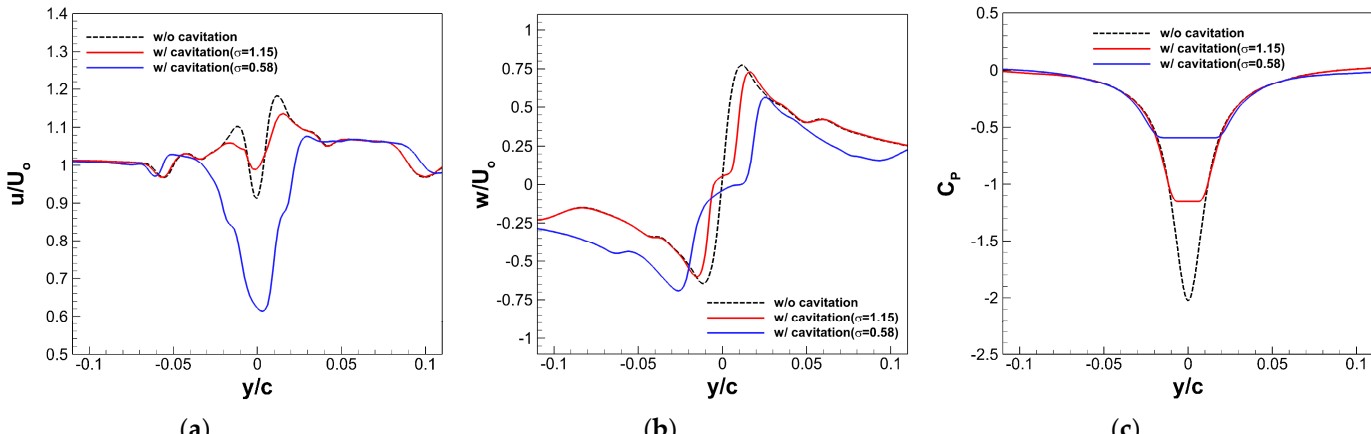

**Figure 14.** Pressure, axial and vertical profiles at x/c = 1.0 for different flow conditions: (**a**) axial velocity; (**b**) vertical velocity; (**c**) pressure coefficient.

### 3.4. Scaling of Tip Vortex Cavitation Inception

This section shows the numerical deduction of a scaling law for the tip vortex cavitation inception for the current elliptic planform foil with the NACA $66_2$-415 section. The tip vortex flow was simulated at different Reynolds numbers: $5.0 \times 10^4$, $1.0 \times 10^5$, $3.0 \times 10^5$, $5.3 \times 10^5$, $7.0 \times 10^5$, $1.0 \times 10^6$ and $1.2 \times 10^6$. In these computations, the cavitation number $\sigma$ is increased until the tip vortex cavitation was difficult to find around the tip of the foil at each Reynolds number. To compare the deduced cavitation index using the minimum pressure coefficients, the wetted flows at the above same Reynolds numbers were simulated.

Figure 15 shows the pressure coefficients along the vortex core at different Reynolds numbers. As the Reynolds number increased, the minimum pressure decreased, and its location moved slightly downstream from the tip of the foil. This trend was because the tip vortex gained more strength as the vortex sheet rolled up to the tip vortex at higher Reynolds number.

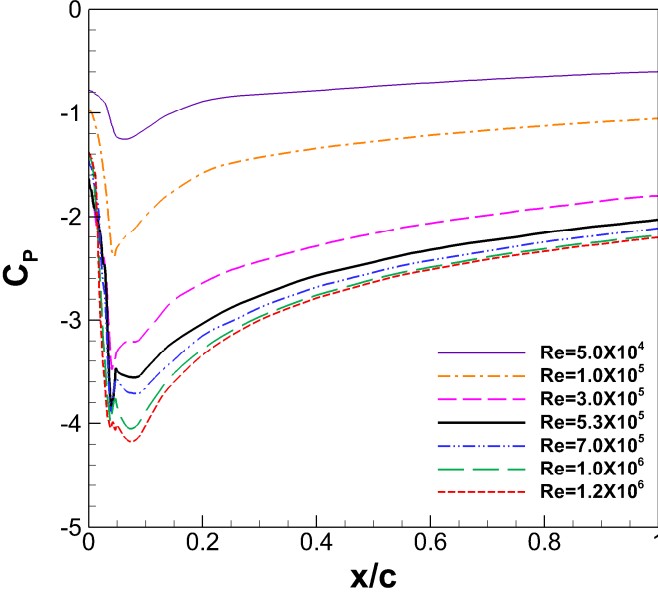

**Figure 15.** Pressure coefficient profiles along the vortex core at different Reynolds numbers.

Figure 16 shows the time histories of the cavitation volumes ($V_{cav}$) predicted with respect to the variation in cavitation number at the following selected Reynolds numbers,

$R_e = 5.0 \times 10^4$, $1.0 \times 10^5$, $3.0 \times 10^5$ and $1.0 \times 10^6$. The simulation time was 1 s, and it is seen that the volume change of the tip vortex cavitation occurring under all flow conditions converged after 0.5 s.

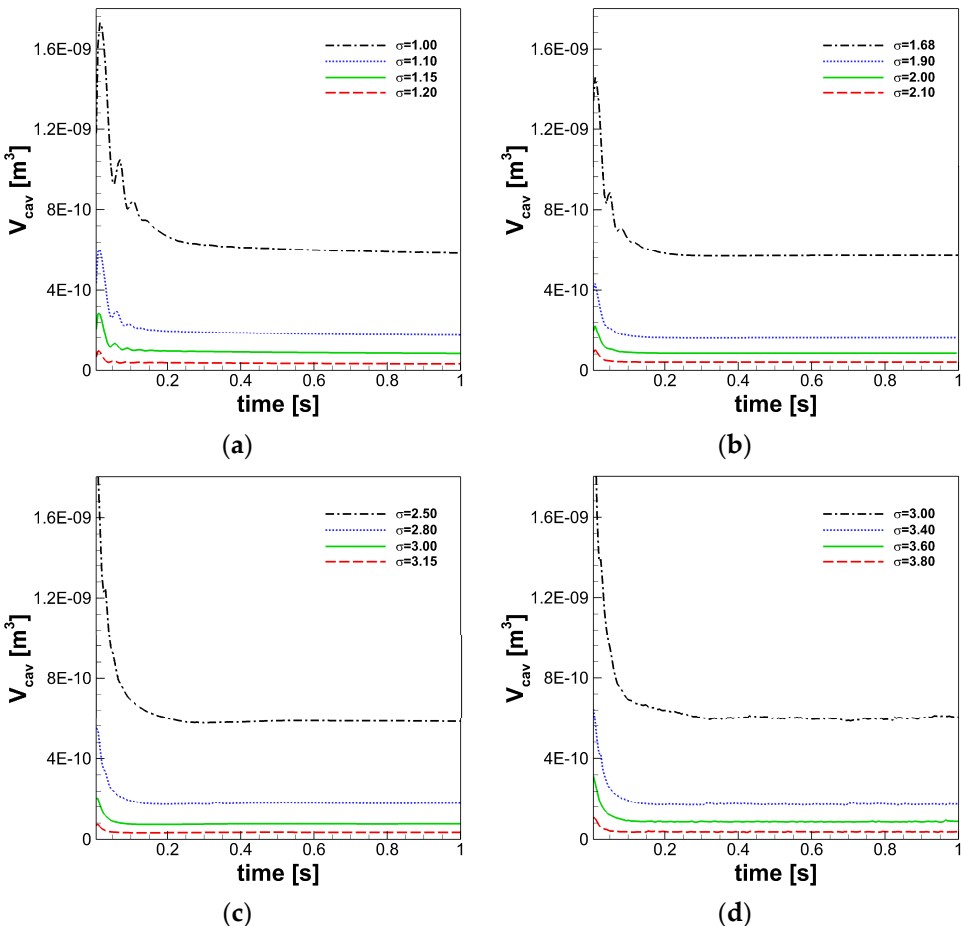

**Figure 16.** Time histories of the cavitation volume at different Reynolds numbers: (**a**) $R_e = 5.0 \times 10^4$; (**b**) $R_e = 1.0 \times 10^5$; (**c**) $R_e = 3.0 \times 10^5$; (**d**) $R_e = 1.0 \times 10^6$.

In Figure 17, the variations in cavitation amount with respect to the cavitation number at different Reynolds numbers are compared. Although the cavitation numbers at each Reynolds number condition were different, they were selected to roughly compare the generated tip vortex cavitation within a similar volume at different Reynolds numbers. It was difficult to accurately estimate the cavitation volume generated under a flow condition which was judged as the cavitation inception point in the experiment. Since the cavitation volume detectable at an inception number was very small, in this study, the cavitation inception number was assumed to be the point where the cavitation volume was nearly zero. The cavitation inception numbers for each Reynolds number, as shown in the figure, were obtained by the extrapolation of the numerical results.

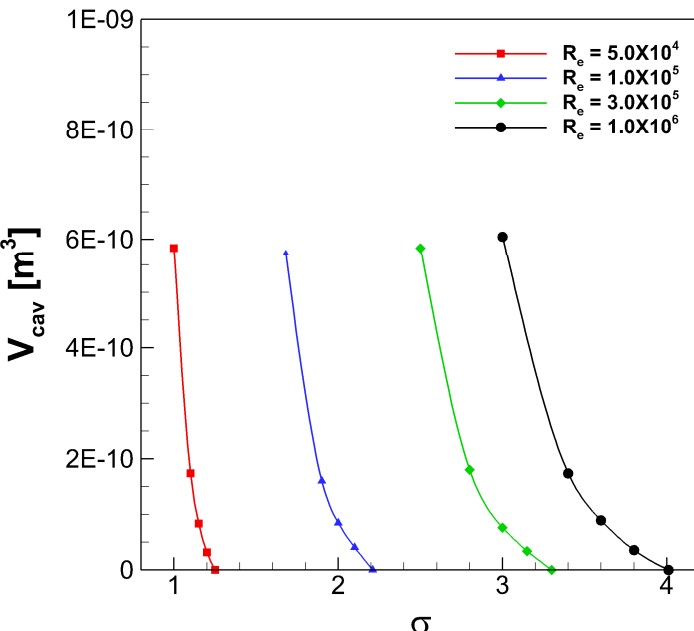

**Figure 17.** Cavitation volume changes with the variation of the Reynolds number and cavitation number.

Figure 18 shows the iso-surfaces of the predicted tip vortex cavitation in the order of cavitation number from small to large values at the given Reynolds numbers mentioned in Figure 17. In the figures, the iso-surfaces of the tip vortex cavitation predicted by a roughly similar volume are compared between different Reynolds numbers. This shows that the tip vortex cavitation became slightly thinner and longer as the Reynolds number increased. Since the present cavitation simulation was included in a weak water problem with a sufficiently large supply of nuclei, it can be found that the cavitation inceptions occurred at a small distance off the tip of the hydrofoil as discussed by Arndt [5]. Unlike the behavior of the minimum pressure shown in Figure 15, the starting point of the tip vortex cavitation moved closer to the tip of the foil as the Reynolds number increased. Mostly, the tip vortex cavitation started to occur at the region preceding the location of the minimum pressure for all given cavitation numbers. For this reason, the variation of the pressure according to the Reynolds number in this region as shown in Figure 15 deter-mined the starting point of cavitation occurrence.

Figure 19 compares the predicted cavitation inception numbers for the variation of the Reynolds number by wetted flow simulations and cavitation simulations. The cavitation inception numbers predicted by the wetted flow simulations showed slightly higher values than those estimated by the cavitation simulations. If the minimum volume observable at a cavitation inception point is used, the difference between the two methods can be slightly larger. The slope of the cavitation index change decreased with increase in the Reynolds number.

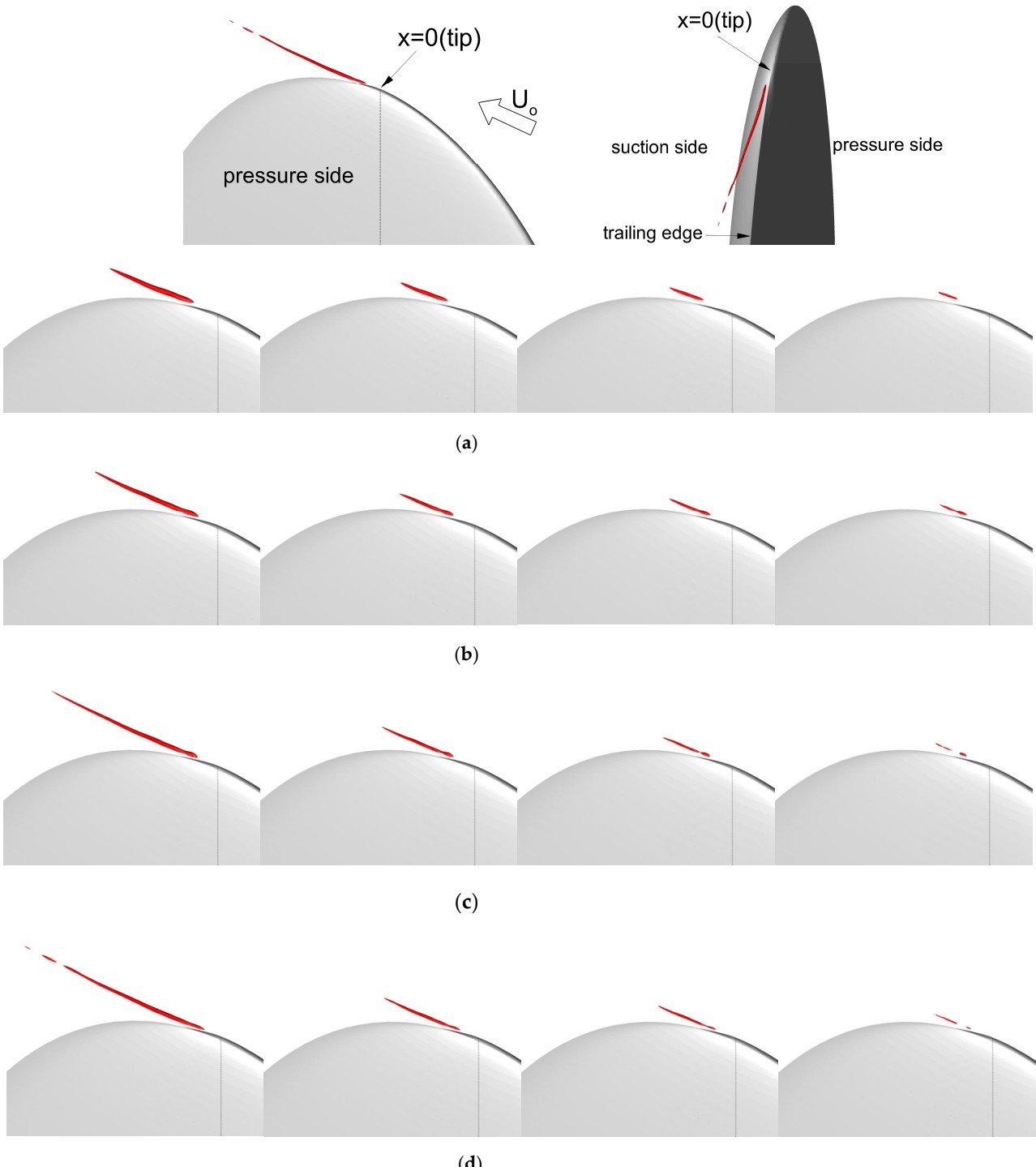

**Figure 18.** Tip vortex cavitations predicted at different Reynolds numbers and cavitation numbers: (**a**) $R_e = 5.0 \times 10^4$, $\sigma = 1.0$, 1.1, 1.15 and 1.20; (**b**) $R_e = 1.0 \times 10^5$, $\sigma = 1.68$, 1.90, 2.0 and 2.1; (**c**) $R_e = 3.0 \times 10^5$, $\sigma = 2.5$, 2.8, 3.0 and 3.15; (**d**) $R_e = 1.0 \times 10^6$, $\sigma = 3.0$, 3.4, 3.6 and 3.8.

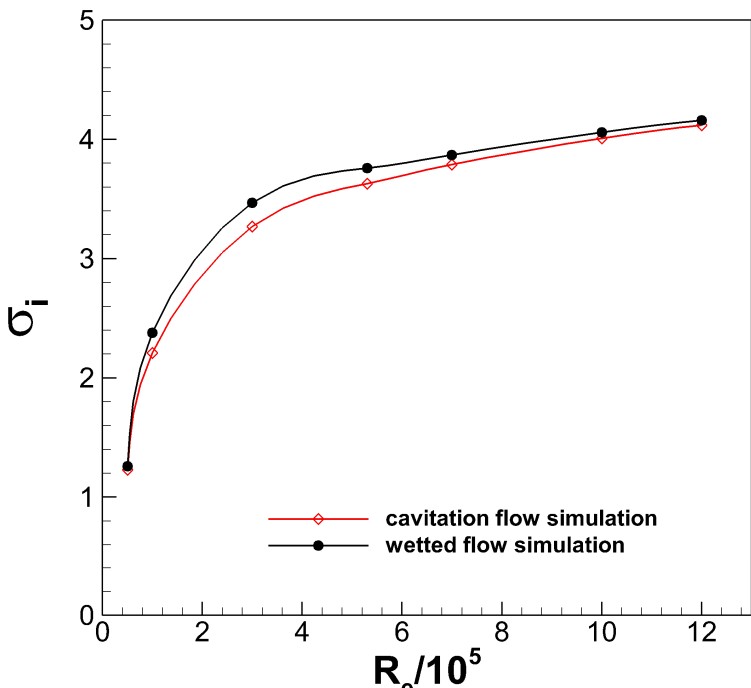

**Figure 19.** Cavitation inception index with the variation of the Reynolds number.

In Figure 20, the predicted cavitation inception numbers are compared over a range of lift coefficient with the cavitation index measured in weak water at a constant velocity, $R_e = 1.0 \times 10^5$ [8]. The experimental curve was obtained by adjusting the measured value of cavitation inception number by the amount of measured value of tension [42]. Although the lift coefficient and cavitation index were obtained at different Reynolds number and angles of attack, the current numerical result showed a reasonable trend with the experiment.

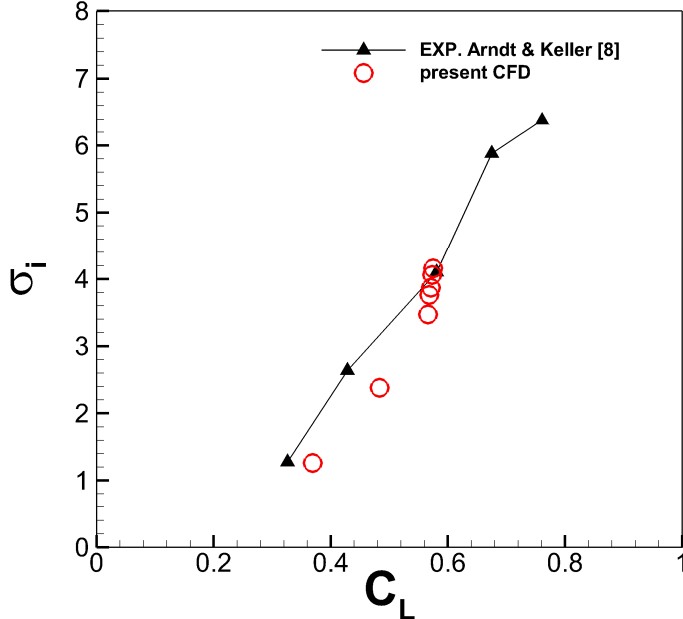

**Figure 20.** Cavitation inception index in weak water with the variation of the lift coefficient.

The cavitation inception numbers obtained from the numerical analysis were further compared with those calculated from the empirical formulas for cavitation inception scaling

derived based on the experimental results (see Table 2). The first scaling equation was McCormick's original scaling law to relate inception to the angle of attack as follows:

$$\sigma_i = \alpha^{1.4} R_e^m \tag{13}$$

In Equation (13), the Reynolds number exponent $m$ was suggested by McCormick [22] to be approximately 0.35. The next scaling equation was Arndt's law [5,8] that uses the lift coefficient of the foil for the Reynolds scaling as follows:

$$\sigma_i = 0.068 C_L^2 R_e^n \tag{14}$$

The constant of 0.068 in Equation (14) is named as $K$, and it varies owing to a secondary effect caused by differences in vortex roll-up for different blade sections and can be determined from cavitation experiments [5]. The Reynolds number exponent $n$ is generally accepted to be approximately 0.4. In the Table 2, both the cavitation scaling equations with the original exponent of the Reynolds number overestimated the cavitation inception index when compared to the present numerical results. This is possibly because of certain numerical errors and a pending problem of imperfection in the scaling of vortex cavitation inception from model tests to full scale conditions. In the current study, new Reynolds number exponents for the two scaling laws were suggested based on the predicted cavitation index. Each Reynolds number's exponent m and n varied with the Reynolds number and tended to decrease lower continuously than the classic constants of 0.35 and 0.4.

**Table 2.** Cavitation inception numbers and scaling index variations in weak water.

| $R_e$ | $C_L$ (wet. Simulation) | $\sigma_i$ (cav. Simulation) | $\sigma_i$ ($C_{p,min}$) | $\sigma_i$ (McCormick' Eq.) | $\sigma_i$ (Arndt's Eq.) | Exponent $m$ | Exponent $n$ |
|---|---|---|---|---|---|---|---|
| $5.0 \times 10^4$ | 0.369 | 1.23 | 1.26 | 2.33 | 0.70 | 0.291 | 0.452 |
| $1.0 \times 10^5$ | 0.484 | 2.21 | 2.38 | 2.96 | 1.59 | 0.325 | 0.429 |
| $3.0 \times 10^5$ | 0.566 | 3.27 | 3.47 | 4.35 | 3.38 | 0.327 | 0.398 |
| $5.3 \times 10^5$ | 0.568 | 3.63 | 3.76 | 5.31 | 4.28 | 0.321 | 0.388 |
| $7.0 \times 10^5$ | 0.571 | 3.79 | 3.87 | 5.86 | 4.82 | 0.318 | 0.382 |
| $1.0 \times 10^6$ | 0.573 | 4.01 | 4.06 | 6.63 | 5.61 | 0.314 | 0.376 |
| $1.2 \times 10^6$ | 0.575 | 4.12 | 4.16 | 7.07 | 6.07 | 0.312 | 0.373 |

As shown in Figure 21, when extrapolated over the range of the Reynolds number used in the current simulations, the suggested scaling exponents behaved as a function of the Reynolds number and asymptotically converge as the Reynolds number increased. The extracted empirical formulas of the exponents over the asymptotic region larger than the Reynolds number of $R_e = 3.0 \times 10^5$ are as follows:

$$m = -0.012 \ln(R_e) + 0.48 \tag{15}$$

$$n = -0.018 \ln(R_e) + 0.63 \tag{16}$$

Table 3 shows the values of the current two scaling exponents in logarithmic form for the Reynolds numbers in the range $5.0 \times 10^5$–$1.0 \times 10^9$. The modified McCormick' exponent m was in the range 0.32–0.23 and the Arndt's exponent $n$ in the range 0.39–0.26 for the low to high Reynolds numbers. Past studies reported that the modified McCormick' scaling exponent was in the range 0.35–0.15 for the turbulent flow regime [24,33,35]. It should be noted that the present study on the cavitation scaling show how the scaling law can be changed with the Reynolds number except for the laminar flow regime. In addition, it is quite useful that a proper scaling exponent for a given Reynolds number can be selected by using Equations (15) and (16).

**Table 3.** Suggested scaling exponents for five main Reynolds numbers.

| $R_e$ | $5.0 \times 10^5$ | $1.0 \times 10^6$ | $1.0 \times 10^7$ | $1.0 \times 10^8$ | $1.0 \times 10^9$ |
|---|---|---|---|---|---|
| $m$ | 0.32 | 0.31 | 0.29 | 0.26 | 0.23 |
| $n$ | 0.39 | 0.38 | 0.34 | 0.30 | 0.26 |

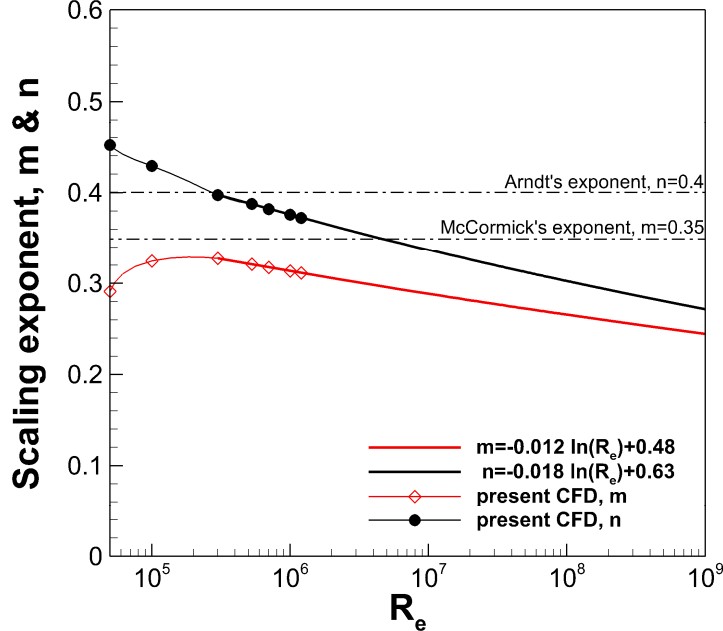

**Figure 21.** Suggested scaling exponents in weak water as a function of the Reynolds number for different scaling laws.

## 4. Conclusions

In the current research, the tip vortex cavitation inception on a hydrofoil was numerically estimated under weak water condition and new scaling laws with variable scaling exponent were further suggested for cavitation inception index. Numerical simulations of the wetted and cavitating flows around an elliptic planform foil were carried out using the RANS approach with a Eulerian cavitation model. The influence of the turbulence models on the steep velocity field inside the vortex was larger than the grid dependence. The Reynolds stress model for the finest grid showed a good agreement with the velocity field measured across the tip vortex. In addition, this model also showed a good correlation with the experimental results of boundary layer characteristics in the tip region of the foil surface where the tip vortex and cavitation developed.

A comparison between the wetted and cavitating flows revealed that the presence of the cavitation in the tip vortex core, which is a phase change accompanying the mass transfer phenomenon, decreased the pressure and velocity gradients resulting in a decrease in the vortex rotation. The minimum pressure near the foil tip decreased as the Reynolds number increased, and its location slightly moved downstream from the tip of the foil. This was the trend because the tip vortex gained more strength as the vortex sheet rolled up into the tip vortex at higher Reynolds numbers.

Since the present cavitation simulation was included in a weak water problem with a sufficiently large supply of nuclei, the tip vortex cavitation started at a small distance off the tip of the hydrofoil. The starting point of the cavitation moved closer to the tip of the foil as the Reynolds number increases. In the comparison of the cavitation inception prediction, the wetted and cavitating flow simulation approaches showed a reasonable agreement between the simulation and experimental cavitation index values.

Finally, the present study has suggested new empirical formulas as a function of the Reynolds number substitutable for the two classic constant scaling exponents. These

formulas were derived by extrapolation based on the current numerical cavitation index and show an asymptotic converge as the Reynolds number increases. It was found that the new McCormick' exponent $m$ was in the range 0.32–0.23 and the Arndt's exponent $n$ in the range 0.39–0.26 for the Reynolds number in the range $5.0 \times 10^5$–$1.0 \times 10^9$. To conclude, the present study showed how the scaling law changes with the Reynolds number and provided quite useful formulas to properly determine the scaling exponent for a given Reynolds number. In the future, this suggested model is expected to be useful in full scale applications.

**Author Contributions:** Conceptualization, I.P., B.P. and H.S.; methodology, I.P. and J.K.; software, J.K.; validation, J.K. and I.P.; investigation, I.P. and J.K.; writing—original draft preparation, I.P.; writing—review and editing, I.P., B.P. and H.S.; supervision, B.P. and H.S.; project administration, B.P. and H.S. All authors have read and agreed to the published version of the manuscript.

**Funding:** This research was funded by Ministry of Trade, Industry and Energy and Defense Acquisition Program Administration and Agency for Defense Development.

**Acknowledgments:** This study was supported by grants from the dual use technology project, 'Three-dimensionally curved twisted-rudder technology' of KRISO (PNS3840) and 'Future Submarine Low Noise Propeller Research Laboratory' of KRISO (PGS4261) funded by Defense Acquisition Program Administration and Agency for Defense Development.

**Conflicts of Interest:** The funders had no role in the design of the study; in the collection, analyses or interpretation of data; in the writing of the manuscript or in the decision to publish the results.

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
