# Peer review of "Numerical Study on Tip Vortex Cavitation Inception on a Foil"

_applsci, doi:10.3390/app11167332_

Round 1

Reviewer 1 Report

This is an interesting numerical study to help better understand the mechanism of cavitation. The authors wisely applied numerical and scale modeling to this complex cavitation problem by carefully conducting numerical simulations and then applied those results to improve empirical scaling laws.  This is a smart approach to this complex problem.  This reviewer can recommend publication and offers the following two minor points. (1) It will help readers if the authors can summarize their lengthy conclusions by listing key findings without losing the value, e.g., numerate each of these key findings.  (2) This reviewer recommends the following reference for the authors’ interest associated with their numerical and scale modeling approach. K. Saito and F.A. Williams, “Scale modeling in the age of highspeed computation,” Progress in Scale Modeling, Volume II: 1-16, Springer, 2014.

Author Response

This is an interesting numerical study to help better understand the mechanism of cavitation. The authors wisely applied numerical and scale modeling to this complex cavitation problem by carefully conducting numerical simulations and then applied those results to improve empirical scaling laws.  This is a smart approach to this complex problem.  This reviewer can recommend publication and offers the following two minor points.

Point 1:  It will help readers if the authors can summarize their lengthy conclusions by listing key findings without losing the value, e.g., numerate each of these key findings.

Response 1: As pointed out by the reviewer, the conclusion has been more concisely summarized.

Point 2: This reviewer recommends the following reference for the authors’ interest associated with their numerical and scale modeling approach. K. Saito and F.A. Williams, “Scale modeling in the age of highspeed computation,” Progress in Scale Modeling, Volume II: 1-16, Springer, 2014.

Response 1: I expect that the reference you recommended will provide new insight into the analogy of the cavitation inception problem. In the future, we will conduct advanced research that reflects this.

Reviewer 2 Report

This is very interesting manuscript advancing the knowledge of cavitation. Before this manuscript can be accepted for publications, authors needs to add paragraph describing comparison with experimental results in way more details. The authros did not have any experimental fluid dynamics results created by themself, but they rely on reference 7.  Authors will need to paraprhase, with appropriate citations, summary results and experimental setup from Ref 7 (and any other they used to), and clearly explains how the experimental conditions directly correlate with their model. Are the geometry and flow conditions the same ? 

The article pedagogical value will be way more improved, and thereby future citations will be higher, if the authors expands the introduction section and describes in greater details some of the underlying assumptions, for example explain how does the Reylonds tensor applies for the cavitation and explain its concept. 

The authors should make the model files available to readers as an attachment to the manuscript. 

This manuscript should be accepted for publication after the minor revisions, as per above, no additional review required.

Author Response

This is very interesting manuscript advancing the knowledge of cavitation.

Point 1:  Before this manuscript can be accepted for publications, authors needs to add paragraph describing comparison with experimental results in way more details. The authros did not have any experimental fluid dynamics results created by themself, but they rely on reference 7.  Authors will need to paraprhase, with appropriate citations, summary results and experimental setup from Ref 7 (and any other they used to), and clearly explains how the experimental conditions directly correlate with their model. Are the geometry and flow conditions the same ?

Response 1: We would like to assure the reviewer that the geometry of the foil and the experimental conditions described in Arndt et al. [7] and Maines & Arndt [40] that we quoted and compared were precisely reproduced in the current numerical analysis as they are. You can see that the shape of the wing is NACA 662-415 in their paper, and we also used the same NACA 662-415. Figure1 shows the shape and main dimensions of the foil used in their experiments, and Figure 2 shows the computational domain reflecting the size of the water tunnel, the experimental facility. To reflect comments from other reviewer, we have revised a sentence that could cause misunderstandings about the angle of attack. Figure 9, showing the results of the NACA 16-020 foil, was included to be used as explanatory data for Figure 10. However, this seems to cause a misunderstanding that the comparison between the current numerical results for and the experimental data was carried out for different foils, i.e. NACA 662-415 foil and NACA 16-020 foil. Therefore, we have modified Figure 9 by adding a supplementary explanation.

In the case of Inception analysis, the geometry of the foil is the same, and the conditions of angle of attack and flow speed are different. However, the cavitation inception index should show the same value for a given lift force. From this point of view, comparisons were made with experiments [40]

Reviewer 3 Report

This paper investigates various numerical schemes for modeling tip vortex cavitation. This work is very important, and can reduce the search for numerical tools to be used in future engineering designs. I do have two major concerns which may be (or not) easily clarified.

a) The actual flow described in Fig. 2 is probably laminar to begin with! It may be mostly laminar on the pressure side and transition at a certain distance on the suction side. Now, it is clear that direct simulation would be a real challenge here – so the authors must do some ‘hand waving’ to clarify this. To help them, I’d suggest to state that in order to simplify the ‘engineering’ modeling – let us assume the flow is turbulent everywhere, but we know it is NOT.

On another note, the paper doesn’t discuss the surface friction! Only the pressure distribution is sought. So why not just solve the Euler equations with some thin BL model? Can the author explain the effect of their fully turbulent model on the surface friction (probably not an easy task).

b) The angle of attack discussion (particularly is section 3) is problematic! Any aerodynamic text will tell that the zero-lift-angle is a negative angle so the lift can be calculated as
?L= ?L∝(∝ −∝L0)= ?L∝(12 − (−2.5))
So, simulation must be done at 12 deg and not at 9.5!!!

Also, there is no mention of the lift slope, so why discuss the zero lift angle???
This is a concern because when comparing with experiments – sometime data at 9.5 is used and some time at 12 deg. This is not acceptable!

Some other comments:

In the ‘Abstract,’ the wingtip model is described as an ‘elliptical hydrofoil.’ It is really an elliptical shaped planview of a wingtip. Just to clarify this: in wing theory, an elliptic wing has elliptic loading. For an untwisted planform the top view of the wing is also elliptic.
Plus, there is no mention of twist – is it zero?

Just to have a positive comment: In the ‘Introduction,’ the authors clearly state that they are trying to test different approaches and document which may be the best for wing tip cavitation modeling.

Please define cavitation index the first time it is mentioned.

In Section 2; “ Numerical approach’: please see my first concern.

In Section 2.2; How the bubble radius R is estimated? Is it a uniform radius, a distributed function or???
Section 3; ‘Numerical results’

This discussion is problematic – because results were compared with the wrong experimental data! Is it 9.5 deg or 13 deg ??? See my second concern up front.

Fig 10 pressure side must be easy to calculate (could be laminar) – why is the surprise?

Section 3.3; was the Eulerian cavitation model used here? Please explain.

Fig 12 (d); What is the vertical line – please explain.
At this point, I’d like to see a chordwise pressure distribution (upper/lower) and perhaps see where transition would be, if modeled correctly.

Fig 15; Please explain if Vortex core is vortex centerline?

Rest of the results show important features of the flow (once we accepted the solution method). If previous concerns are properly addressed, this could be a valuable contribution.

Author Response

Point 1: The article pedagogical value will be way more improved, and thereby future citations will be higher, if the authors expands the introduction section and describes in greater details some of the underlying assumptions, for example explain how does the Reylonds tensor applies for the cavitation and explain its concept.  

Response 1: As noted by the reviewer, we also agree that this issue is very interesting. However, this work did not focus on this problem, and just used a general solution procedure. In this study, the governing equations were solved through a segregated solution approach. After solving the continuity equation and the momentum conservation equations, the turbulence equations are solved, where the Reynolds stress term is solved using a numerical model or separate equations for turbulence closure. At this time, the velocity field is newly updated according to the solution of the turbulence equations, and cavitation is calculated by using a transport equation based on this velocity field as shown in Equation (11). That is, the effects of the Reynolds stress tensor are expressed as a velocity field. In other word, the effect of the Reynolds tensor term on cavitation is transmitted through the updated velocity field. The numerical approach described above is not a special method presented by our work. Therefore, rather than presenting a separate additional detailed explanation in the Introduction, we believe that Sections 2.2 and 2.3 can provide a clear and sufficient description of the relation between the Reynolds stress term and cavitation modelling.

Point 2: The authors should make the model files available to readers as an attachment to the manuscript.

Response 2: We are so sorry, we don’t catch well the meaning of the sentence “make the model files available to readers”

This manuscript should be accepted for publication after the minor revisions, as per above, no additional review required.

This paper investigates various numerical schemes for modeling tip vortex cavitation. This work is very important, and can reduce the search for numerical tools to be used in future engineering designs. I do have two major concerns which may be (or not) easily clarified.

Point 3: a) The actual flow described in Fig. 2 is probably laminar to begin with! It may be mostly laminar on the pressure side and transition at a certain distance on the suction side. Now, it is clear that direct simulation would be a real challenge here – so the authors must do some ‘hand waving’ to clarify this. To help them, I’d suggest to state that in order to simplify the ‘engineering’ modeling – let us assume the flow is turbulent everywhere, but we know it is NOT. On another note, the paper doesn’t discuss the surface friction! Only the pressure distribution is sought. So why not just solve the Euler equations with some thin BL model? Can the author explain the effect of their fully turbulent model on the surface friction (probably not an easy task).

Response 3: It is believed that the reviewer summarized the main flow phenomena of this study very well. Also, as mentioned by the reviewer, we know that transition flow (laminar to turbulent) is a very important issue in numerical modelling other than DNS (direct numerical simulation). However, prior to this study, the authors confirmed the ability of the Reynolds stress model (RSM) to capture the 'laminarization' phenomenon in vortex flows, in which the turbulent kinetic energy is minimized. Contrary to the reviewer' comments, this paper deals with surface friction named as wall shear stress in Figure 10 as well as the pressure (Figure 11). Comparing the results with the experimental data, it can be found that the RSM captures the vortex separation on the suction side well. Although turbulent flow is assumed, it is seen that the RSM model reproduces the main flow phenomenon on the suction side mentioned by the reviewer well.

Point 4: b) The angle of attack discussion (particularly is section 3) is problematic! Any aerodynamic text will tell that the zero-lift-angle is a negative angle so the lift can be calculated as

CL= CL∝(∝ −∝L0)= CL∝(12 − (−2.5))

So, simulation must be done at 12 deg and not at 9.5!!!

Also, there is no mention of the lift slope, so why discuss the zero lift angle???

This is a concern because when comparing with experiments – sometime data at 9.5 is used and some time at 12 deg. This is not acceptable!

Response 4: First above all, we would like to say that the angles of attack are not different from the experimental conditions. However, we are sorry that the angle of attack mentioned in the paper caused confusion. In the revised paper, we have corrected it as follows:

“Non-cavitating flows are computed for the foil with the effective angle of attack  αe=12° at the Reynolds number of 5.2x105, where the corresponding flow speed is Uo =6.5m/s. It should be noted that the practical angle of attack  becomes α= 9.5° and the zero-lift angle of attack αo= -2.5° as mentioned in Arndt et al. [7].”

In the Arndt et al. [7], the expression of the effective angle of attack is written as follows: (α-αo )= 12° for wetted flow; (α-αo ) = 9.5° for cavitating flow, where  αo = -2.5°.

The zero-lift angle of attack is an experimentally given value. In this study, the experimental values were used without lift-slop computation to match the experimental conditions [7].

Some other comments:

Point 5: In the ‘Abstract,’ the wingtip model is described as an ‘elliptical hydrofoil.’ It is really an elliptical shaped planview of a wingtip. Just to clarify this: in wing theory, an elliptic wing has elliptic loading. For an untwisted planform the top view of the wing is also elliptic.

Response 5: As pointed out by the reviewer, the name of the foil has been corrected as elliptic planform foil.

Point 6: Plus, there is no mention of twist – is it zero?

Response 6: As shown in Figure 1, this foil do not have a twist.

Point 7: Just to have a positive comment: In the ‘Introduction,’ the authors clearly state that they are trying to test different approaches and document which may be the best for wing tip cavitation modeling.

Response 7: We also hope that the Introduction will give potential readers a better understanding.

Point 8: Please define cavitation index the first time it is mentioned.

Response 8: As pointed out by the reviewer, cavitation index has been defined the first time it is mentioned, which can be found in lines 238 in the page 6.

Point 9: In Section 2; “ Numerical approach’: please see my first concern. In Section 2.2; How the bubble radius R is estimated? Is it a uniform radius, a distributed function or???

Response 9: In this numerical model based on an Eulerian approach, the bubble radius is calculated by using the volume fraction distribution and the Equation (7) given in Section 2.2 as follows:

Point 10: Section 3; ‘Numerical results’

This discussion is problematic – because results were compared with the wrong experimental data! Is it 9.5 deg or 13 deg ??? See my second concern up front.

Response 10: As shown in the answers to the Point 4, we would like to assure you that the angles of attack are not different from the experimental conditions. This misunderstanding has been rectified in the revised paper. Please, see again Response 4.

Point 11: Fig 10 pressure side must be easy to calculate (could be laminar) – why is the surprise?

Response 11: As the reviewer mentioned, in conclusion, there is no significant flow change in the pressure side, but the purpose is to provide the results of the flow change in the pressure surface according to turbulence model for the readers.

Point 12: Section 3.3; was the Eulerian cavitation model used here? Please explain.

Response 12: Yes, that's right. This is the result of using the Eulerian cavitation model described in the Section 2.2 as it is.

Point 13: Fig 12 (d); What is the vertical line – please explain.

Response 13: The vertical line is the boundary of sheet cavitation that covers the foil surface. The same result can be found in Frank et al. [43] which is newly added in the current reference. The presumed reasons for this difference have been written in the paper as follows:

“On the other hand, in the current numerical analysis result, it is seen that sheet cavitation covers the surface of the foil. The same trend can be found in the numerical results of Frank et al. [43]. This might be due to the surface condition of the wing in the experiment and the nuclei condition in the cavitation tunnel, which was not considered in numerical analysis”

Point 14: At this point, I’d like to see a chordwise pressure distribution (upper/lower) and perhaps see where transition would be, if modeled correctly.

Response 14: As requested by the reviewer, the pressure profiles at four span locations are provided below for reference.

Point 15: Fig 15; Please explain if Vortex core is vortex centerline?

Response 15: Yes, that's right. The pressure in the figure is the value obtained along the centerline tracing the vortex core.

Rest of the results show important features of the flow (once we accepted the solution method). If previous concerns are properly addressed, this could be a valuable contribution.
